# Perceived Impact of the COVID-19 Pandemic on Playing Golf: A Qualitative Content Analysis Study

**David Jungwirth** [1,†] , **Susanne Gahbauer** [2,†] **and Daniela Haluza** [1,*,†]

1  Center for Public Health, Department of Environmental Health, Medical University of Vienna, Kinderspitalgasse 15, A-1090 Vienna, Austria
2  Center for Public Health, Department of Social Medicine, Medical University of Vienna, A-1090 Vienna, Austria
*  Correspondence: daniela.haluza@meduniwien.ac.at
†  These authors contributed equally to this work.

**Abstract:** Golf is a very popular outdoor sport played worldwide by people from various socio-economic backgrounds. During the COVID-19 pandemic, lockdowns and quarantine restrictions led to closures of indoor and outdoor sport facilities and thus also affected the access to golf courses. This study aimed at elucidating perceptions of golfers regarding the impact of the crisis on their sport. We surveyed a sample of golfers in German-speaking countries (primarily Austria and Germany) from March to June 2021. A content analysis on golfers' responses ($n$ = 923) to an open-ended question on their feelings and engagement in alternative ways of physical activity in light of closed sport facilities was performed. More than 23% of the comments concerned negative feelings on how the pandemic impacted the surveyed golfers, especially due to reluctance to accept the closure of outdoor sports facilities. Almost all participants stated to perform alternative outdoor sports, mostly hiking, biking, walking, and running. In times of a pandemic, public spaces should be designed to allow for safe physical activity to maintain a mentally and physically fit population. In this context, policy makers should provide hygiene concepts that allow for minimal disturbance of sport routines, especially in regard to outdoor sports such as golf.

**Keywords:** health promotion; exercise; public health; pandemic; nature contact; content analysis; online survey

## 1. Introduction

Regular physical activity is key for promoting health and well-being, and preventing non-communicable disease, often associated with a sedentary lifestyle [1,2]. Golf is an organized sport with well-received opportunities for regular social contacts and community engagement, which are known to play an important role in health outcomes [3]. Evidently, the low to moderate intensity level physical effort associated with playing golf contributes to the general fitness in terms of cardio-respiratory and metabolic health, bone density, and balance of the athletes [4,5]. In a large Swedish cohort study on effects of leisure time physical activity, active golfers showed a 40% reduced mortality rate, resulting in a higher life expectancy of about five years, compared to the general population [6]. Additionally, the individually adjustable intensity level allows golfers of all ages and physical abilities to competitively take part in the game. This is translated into the high popularity of golf for people being in their fifties and even older [3,7].

In March 2020, governments reacted to the emergence of the COVID-19 pandemic with massive restrictions on business, education, travel, sports, and socio-cultural events [8]. The lockdowns and social distancing measures resulted in the longer-term closure of indoor and outdoor sport facilities such as fitness centers, sport clubs, and also golf courses in all regions of the globe [9,10]. Recently, a rising number of international studies reported on the adverse impact of the COVID-19 restrictions, with physical activity substantially

decreasing, and screen-related sedentary behavior increasing [11–13]. Additionally, quality of life, life satisfaction, mental health, and well-being declined, mirroring the psychosocial strain experienced during lockdowns and other restrictions, limiting social contacts and physical activity alike [12,14–18].

In line with previous studies on health and well-being benefits of exposure to outdoor nature, people were eventually allowed to exercise outdoors during COVID-19 mobility restrictions [19–22]. Vast scientific evidence on the importance of public urban green and blue spaces and peri-urban nature as alternatives for closed indoor and outdoor sports facilities is accumulating [17,22–24]. Grigoletto et al. found that during the COVID-19 pandemic, people visited urban parks in order to relax and to engage in physical activity in an Italian sample [25]. As a result, green spaces and inner-city parks were heavily used for recreational purposes and became overcrowded, as shown in a study from Vienna [23]. Nevertheless, golf is played on a wide-spread course with a characteristic pattern of holes and structures. So, due to these extremely specific features, golfers are unlikely to find acceptable substitutes neither in a public space nor indoors.

In the last decade, home-based exercising, triggered by the huge investments from the online fitness sector, experienced a global boom [26]. As shown in related studies, COVID-19-related curfews, restrictions, and sports facility closures even supported this trend, also among golfers and other athletes [17,18,27]. Similar to enthusiasts of other popular sports such as tennis or soccer, many golfers watch golf tournaments live or on screen. During the pandemic, more than 70% of U.K. golfers watched golf on TV, and more than 80% engaged in home-based golf-related activities such as practicing golf swings and putting [5]. Notably, these are essential golf practices, but clearly lack other vital aspects typical for the sport of golf such as conducting wide-range shots on target, long-distance walking, social interactions, nature contact, and direct competition with other golfers. Wheatley and Bickerton showed that golf can yield higher levels of life satisfaction compared to similar moderate intensity level sports and leisure activities [28]. In a British study, Sorbie et al. found higher levels of mental health measures such as life satisfaction, sense of belonging, enjoyment, and well-being in golfers compared to the general population [4]. Golf courses re-opened intermittently between the lockdowns, but with strict restrictions including wearing face masks, social distancing rules, closed locker rooms, showers, and clubhouses, which should limit virus transmission, but also hampered social contacts. These might account for the findings that golfers' satisfaction of life significantly improved after golf courses were re-opened in the U.K. after the first lockdown [29].

So far, very little is known on golfers' perceptions, experiences, and reasons for changed behavioral patterns in physical practice during the COVID-19 period, although the sport is very popular among the middle-aged and elderly strata of the population, who especially profit from social contacts and regular exercise to improve cardio-vascular and metabolic health [4,30]. In a recently published cross-sectional online study surveying a large sample of German-speaking golfers, we reported that the pandemic led to a shift of physical activity from indoor to outdoor spaces, given closures of sports facilities, and negatively affected life satisfaction in golfers [27]. So, this study added so-far lacking knowledge on the rapidly increasing amount of scientific literature on the crisis' effects on the specific example of the outdoor sport golf [22,24,30,31]. A closer analysis of responses to an open-ended question collected in this survey among a large sample of golfers could help to understand the reduced levels of life satisfaction by about 24% when comparing ratings before and during the pandemic. To our knowledge, the present study is the first qualitative assessment aimed at exploring circumstances and conditions influencing golfers' reactions to the pandemic-related closures.

## 2. Methods

### 2.1. Study Design and Questionnaire

The present non-representative, cross-sectional online study gathered self-reported data on physical activity and life satisfaction in the context of the COVID-19 pandemic

among German-speaking adult golfers. Details on data collection have been previously published elsewhere [27]. In short, we distributed the online survey link via social networks Facebook and WhatsApp of several golf organizations and golf clubs, as well as personal invitations. The survey was confidential and anonymous, and the respondent could terminate their participation at any time without giving a reason. Ethical approval was granted from the institutional ethical committee of the Medical University of Vienna, Austria, on 1 April 2021, and the study was conducted in full accordance with the guidelines laid out in the Declaration of Helsinki.

The survey was accessible online using the web-based survey tool SoSci Survey from 1 April to 6 June 2021 [32]. The questionnaire collected information on included measures regarding socio-demographic characteristics age (in years), gender (male, female, or diverse), education level (primary, secondary, or tertiary), country of residence (Austria, Germany, or other countries), and golf experience (0–5 years, 6–10 years, or 11 years or longer), and region (urban or rural). In addition to these forced-choice items, the final question of the survey was open-ended and asked participants to describe in their own words what came to their mind with respect to this question: "Due to the COVID-19 pandemic, the golf courses were at least partially closed. We are therefore interested in (1) how you experience this situation and, (2) which alternative sports you practice". Subsequently, qualitative content analysis was used to examine golfers' responses to this question.

### 2.2. Qualitative Data Analysis

Data sets of German-speaking adults that played golf, as indicated by a filter question, and provided free-text comments in the online survey were included in this qualitative analysis. To ensure methodological integrity, we followed the consolidated criteria for reporting qualitative research (COREQ), a 32-item checklist to picture study methods, context of the study, findings, analysis, and interpretations [33]. For qualitative content analysis of the open-ended responses, we used a data-driven inductive approach to code content into themes [30]. Thematic analysis provides a framework for structuring qualitative data by establishing a coding system in which codes are grouped into recurrent themes that are relevant to the research question, allowing for clustering and interpreting complex raw data [31].

All involved researchers had ample experience in performing qualitative analysis. In order to identify themes, a researcher (S.G.) thoroughly examined the responses and noted recurring issues. This researcher then re-examined the responses with these preliminary themes in mind, and either eliminated or verified and defined them. These preliminary themes and their definitions were sent to a second researcher (D.H.), who then examined the responses and noted where themes needed further clarification, and suggested themes to be added or removed. The two researchers discussed and refined the themes until agreement was reached. The final definitions were then recorded in a coding form.

For coding, researchers used their best judgment to determine whether a response fit into a theme, as the definitions were non-exhaustive. At this point, the researchers independently coded the first 25 responses to test interrater reliability and establish agreement using these theme definitions. Agreement was 95%, and the theme definitions were therefore considered as reliable. A third researcher (D.J.), who was not involved in the development of the themes, examined the coding and made a final coding decision on any disagreements. This resulted in the identification of the themes and subthemes where appropriate.

### 2.3. Validity and Reliability

As for qualitative rigor, we used the commonly known evaluative criteria creditability, transferability, dependability, and confirmability [34]. We achieved credibility, referring to confidence in the 'truth' of the findings, by completeness in both data collection and analysis. All coders were familiar with the collected responses by reading through them

several times, ensuring accurate coding. Furthermore, we assured transferability by using direct quotes to exemplify the synthetized results and allowing that the findings might also find applicability in other contexts. Dependability, which aims at minimizing the influence of single analysts on the results, was achieved by installing a third coder who was not involved in the theme development, showing that the findings were consistent and could be repeated. Confirmability was achieved through analyst triangulation involving three researchers. All coders analyzed the verbatim responses, and then validated findings amongst themselves. Additionally, source triangulation was achieved, as responses were collected from golfers of different age and gender in different regions.

*2.4. Statistical Data Analysis*

We performed all statistical analyses using the statistical software SPSS Statistics for Windows, Version 27.0 (Armonk, NY, USA, IBM Corp.). We set statistical significance at $p < 0.05$. We used descriptive statistics to report categorical data as absolute, and relative frequencies and continuous data such as age by mean and standard deviation (SD). We used Mann–Whitney U tests to assess differences in responses of open question non-respondents and open question respondents.

## 3. Results

*3.1. Socio-Demographic Characteristics of the Study Sample*

The final sample included 1115 participants, with more males ($n = 680$, 61.0%) than females ($n = 435$, 39.0%). Of these, the majority of study subjects ($n = 923$, 82.8%) also responded to the open-ended question and it is these participants responses that we examined in the current study (Table 1). Participants who responded to the survey's open question did not differ significantly from non-respondents with regard to education level, country of residence, golf experience, and region ($p > 0.05$ for all variables), as shown by Mann–Whitney U tests. However, open question non-respondents were statistically significantly younger (mean age 55.57, SD 13.09, range 18–85 years) compared to question respondents (mean age 49.83, SD 13.73 years, range 18–85 years, $p = 0.0001$) and of male gender (68% vs. 60%, $p = 0.024$).

**Table 1.** Socio-demographic characteristics of the total study population ($n = 1115$), comparing open question non-respondents ($n = 192$) and open question respondents ($n = 923$).

| | Open Question Non-Respondents ($n = 192$) | | Open Question Respondents ($n = 923$) | | $p$ Value # |
|---|---|---|---|---|---|
| | N | % | N | % | |
| **Gender** | | | | | |
| Female | 61 | 31.8 | 374 | 40.5 | |
| Male | 131 | 68.2 | 549 | 59.5 | 0.024 * |
| **Education level** | | | | | |
| Primary education | 50 | 26.0 | 223 | 24.2 | |
| Secondary education | 75 | 39.1 | 343 | 37.2 | |
| Tertiary education | 59 | 30.7 | 324 | 35.1 | |
| Other | 8 | 4.2 | 33 | 3.6 | 0.398 |
| **Country of residence** | | | | | |
| Austria | 123 | 64.1 | 601 | 65.1 | |
| Germany | 65 | 33.9 | 303 | 32.8 | |
| Other | 4 | 2.1 | 19 | 2.1 | 0.785 |
| **Golf experience** | | | | | |
| 0–5 years | 43 | 22.4 | 147 | 15.9 | |
| 6–10 years | 44 | 22.9 | 221 | 23.9 | |
| 11 years or longer | 105 | 54.7 | 555 | 60.1 | 0.076 |
| **Region** | | | | | |
| Urban | 104 | 54.2 | 487 | 52.8 | |
| Rural | 88 | 45.8 | 436 | 47.2 | 0.723 |
| **Total** | 192 | 100.0 | 923 | 100 | |

Note: # $p$ values from Mann–Whitney U test, * $p < 0.05$.

*3.2. Findings of the Qualitative Data Analysis*

Table 2 shows the results of the qualitative analysis of attitudes towards golf course closures due to COVID-19 and lists alternative sports in the two distinct sections I and II. The themes under section "*I. Attitudes toward golf course closures due to COVID-19*" were ranked following a logical order of occurrence during the crisis, not according to frequency, which was done in section "*II. Alternative sports*".

**Table 2.** Qualitative analysis of attitudes toward golf course closures due to COVID-19 (I) and alternative sports during golf course closures due to COVID-19 (II).

| | N | % |
|---|---|---|
| **I. Attitudes toward golf course closures** | | |
| **Total I** | **534** | **57.85** |
| **1. Negative statements on golf restrictions** | **211** | **39.51** |
| Closure of golf courses is senseless | 125 | 23.41 |
| Impact on emotions, health, and well-being | 48 | 8.99 |
| Infection risk higher in other settings | 19 | 3.56 |
| Anger towards politics | 19 | 3.56 |
| **2. Negative statements on general situation** | **24** | **4.49** |
| **3. Tolerance, acceptance of restrictions** | **37** | **6.93** |
| **4. Golf course closures only temporarily** | **134** | **25.09** |
| **5. Interim terms of use** | **72** | **13.48** |
| **6. Golf as a kind of religion** | **47** | **8.80** |
| **Unspecific comments (non-coded)** | **9** | **1.69** |
| **Sum I** | **534** | **100** |
| **II. Alternative sports (multiple answers)** | | |
| **Total II** | **389** | **42.15** |
| Hiking | 203 | 52.19 |
| Biking | 176 | 45.24 |
| Walking | 159 | 40.87 |
| Running | 154 | 39.59 |
| Indoor sports | 102 | 26.22 |
| Nordic walking | 95 | 24.42 |
| Simulated golf | 85 | 21.85 |
| Yoga, Pilates, Qigong | 55 | 14.14 |
| Skiing, snowboarding | 35 | 9.00 |
| Online fitness | 28 | 7.20 |
| Gardening | 12 | 3.08 |
| Water sports, e.g., swimming, rowing, surfing | 12 | 3.08 |
| Other outdoor sports | 20 | 5.14 |
| Tennis | 9 | 2.31 |
| Cross country skiing | 8 | 2.06 |
| No alternative sports | 37 | 9.51 |
| **Sum II** | **389** | **100** |
| **Total I and II** | **923** | **100** |

In total, 923 participants shared their perceptions in the free-text comment box. Of those, 534 subjects (57.9%) reported material that we coded as "*I. Attitudes toward golf course closures*". Theme 1 "Negative statements on golf restrictions" was the most common theme (39.5%). Given the relevance and complexity of this topic for this study, we further differentiated the quotes encompassed by this theme by four subthemes. From the four subthemes in this theme, "Closure of golf courses is senseless" was most often mentioned (23.4%). Examples for related quotes were "Closed golf courses lack any logic, since exercise

in nature, especially in a non-physical sport, only has positive effects" (male, 48 years), "Blocking the golf courses makes [ . . . ] no sense, as there is no risk for me or for others" (male, 70 years), and "Due to scientific findings of aerosol research, a closure [ . . . ] is completely disproportionate, inappropriate and also not expedient" (male, 64 years).

The second subtheme was related to COVID-19's impact on emotions, health, and well-being (*n* = 48). Examples for quotes were "Hard to bear, because outdoor exercise is extremely important" (female, 55 years), "It is bad, how some people get aggressive when the golf course is closed. I even saw people illegally on the course in April 2020." (female, 54 years), and "I lack an important balance to my everyday work, which has become more stressful due to the pandemic" (female, 48 years).

The third subtheme enclosed the perception that the infection risk in other settings is higher than at the golf course (*n* = 19). Examples for quotes were "I experience this with absolute incomprehension. Especially when I'm trapped in the crowded subway." (female, 50 years), "During the closure I practiced walking, hiking, and biking. I met more people there without a sense of distance than when I am at the golf course!" (female, 51 years), and "[ . . . ] From my point of view, a round of golf with a 4-flight poses little or no risk at all. As long as there are no proper checks on distances and crowds in shopping malls, public transport, etc., it is incomprehensible that the golf courses should be closed" (male, 55 years).

As for the fourth subtheme, some participants expressed their anger towards politics (*n* = 19), enclosing statements also including aspects such as long-term closures and testing. Examples for quotes were "Every single decision-maker, who supports and follows up on these measures after more than six months, must be held legally accountable: personal bankruptcy and restitution to the general public" (male, 35 years) and "A ban on playing golf can therefore only be decided by someone who has absolutely no idea about life" (male, 50 years). A further typical example for quotes in this subtheme is "It's a cheek to close the golf courses because of a pandemic. Since COVID-19 only affects about 2–3% of the population, these measures are not understandable. [ . . . ] If testing by these non-validated and non-diagnostic PCR tests stopped, there would be no pandemic" (female, 57 years).

In contrast to Theme 1 (39.5%), comments relating to Theme 2 "Negative statements on general situation" and Theme 3 "Tolerance, acceptance of restrictions" were only sporadically mentioned (4.5 and 6.9%, respectively).

In Theme 2, participants expressed negative statements on the general situation during COVID-19. Examples for quotes in this theme were "The situation is generally depressing and can only be endured if you at least have no financial worries [ . . . ]" (female, 64 years) and "The pandemic situation is certainly the most bearable for retirees, but due to the long duration it is now also becoming increasingly stressful" (male, 70 years).

As for Theme 3, a minority of surveyed golfers shared feelings in regard to tolerance and acceptance for restrictions. Examples for quotes were "Since I always take a golf break in winter anyway and don't go on golf trips abroad, I hardly noticed any difference compared to previous years" (female, 52 years), "There is such a wide variety of exercise options that the short-term closure of the golf courses is acceptable. [ . . . ]" (female, 55 years), and "My home club was closed for about 8 weeks, so there was a lack of golf, but there are worse things in the pandemic and if lives are saved, that's good" (male, 53 years).

We subsumed statements regarding the notion that golf course closures were only temporary by 134 participants in Theme 4 "Golf course closures only temporarily", which accounted for a fourth of all comments (25.1%). Examples for quotes were "The golf courses I play at were only closed during the first lockdown, and have been open ever since, which is good as I don't engage in any alternative sports" (female, 47 years) and "With the exception of a few weeks, most of the golf courses I know were open, but with restrictions" (male, 72 years).

As for Theme 5 "Interim terms of use", 72 participants shared their perceptions in regard to interim use of golf courses between the lockdowns including ban of tournaments and closure of gastronomy and showers, which resulted in a perceived lack of the social

aspects of golf. Examples for quotes under this theme were "Unfortunately, the available start times are rare. Spontaneity gets lost. No team training, no national tournaments. This eliminates social contacts, fun and team trips or weekend trips due to closed hotels." (female, 55 years) and "Fortunately, [the golf courses] are open at present. Unfortunately, the social part is currently not possible, as there is no training in the group. I really missed the opportunity to go on a golf trip to the south over the winter months." (female, 63 years). Two further typical quotes in this topic were "Clearly, there is no sitting on the terrace AFTER the golf, it is part of the relaxing experience with friends. But still, better than no golf at all" (male, 57 years) and "I miss the unrestricted contacts after the round, the drinks on the terrace with other players, the conversations and the exchange about the round played. [ . . . ]" (female, 42 years).

Further quotes were summarized under the last theme, Theme 6 "Golf as a kind of religion" (*n* = 47). Examples for quotes were "Golf is the last resort" (male, 54 years), "Golf is one of the few bright spots in these almost bleak times" (male, 55 years), "An important part of my life was missing, especially since I also do voluntary work at the golf club." (female, 61 years), and "Thank God the golf courses are open again. It is very pleasant that you can practice the most beautiful thing in the world" (female, 66 years).

Finally, less than 2% of all comments were too unspecific and thus not integrated in one of the themes.

Furthermore, 389 of all participants (42.2%) reported on alternative sport options using multiple answers. The most common sports were outdoor activities, i.e., hiking (52.2%), biking (45.2%), walking (40.9%), and running (39.6%). Indoor sports were mentioned by a fifth of the golfers (26.2%). Simulated golf, such as putting at home or with online golf tutorials, was practiced by 21.9% of the respondents.

## 4. Discussion

This study aimed at exploring perceptions on the impact of the COVID-19 pandemic on playing golf in a large sample of German-speaking golfers. In the present qualitative analysis, we used data from an already published cross-sectional study [27]. As previously reported, we found significantly reduced levels of life satisfaction in the surveyed golfers [27]. So, we were interested in an in-depth analysis of the free-text comments to investigate the perceived challenges and restrictions for the golfers that might have led to mental health reductions. More than 80% of the original questionnaire respondents answered to the respective open-ended survey item. We supposed this to be an unusually high participation rate, highlighting the perceived relevance of pandemic-related concerns to golfers in our sample. Furthermore, many respondents answered the questions in much detail, covering a broad range of emotions, desires, and personal experiences as well as alternative sports, which they practiced.

All six main themes under the umbrella term 'I. attitudes towards golf course closures' identified in our analysis have been mentioned to some degree in public media reports and also golf-related journals and online channels [35–37]. This first section collected the attitudes of golfers, highlighting a broad range of themes within six themes. The first theme comprised negative statements on golf restrictions. Notably, this theme accounted for about 40% of all response codes. Many of these concerned substantial critiques, noting the frustration that access to golf course was constrained, although it is a sport predominantly played outdoors on large areas allowing for keeping adequate social distance in times of COVID-19 restrictions, and thus a limited virus transmission risk [38]. According to their statements, the golfers felt that a vital part of their life was taken from them, at least for a short time, without a reasonable and fair concept from decision makers to restrict similar other outdoor exercise [26].

This notion was also taken up by the second theme, negative statements on the general situation, which clearly was per se also affecting feelings of the golfers. In this theme, about 5% of the participants were especially driven by their notion of mismanagement by decision makers directly contributing to the population´s health and well-being, and some

of the comments were rather emotional and critical. We supposed that the anonymous nature of an online assessment tool was especially valuable for capturing these kinds of feelings of study subjects. The negative effect of COVID-19 restrictions on mental health and health-related habits in all strata of the population across countries worldwide have already been documented by several researchers worldwide [12,15]. People's reluctance toward governmental COVID-19 prevention measures, especially regarding wearing masks and vaccinations, also accompanied with a rise of conspiracy beliefs, scientific skepticism, and general displeasure is well documented [39]. Nevertheless, we did not detect any significant expression related to conspiracy theories, which might root, e.g., in the relative high education level of participants, but could account for the social desirability bias often experienced in online surveys.

Only few participants (about 7%) expressed optimistic feelings related to tolerance and acceptance of restrictions, which formed the third theme. However, many surveyed golfers were glad that the golf courses were only temporarily closed and re-opened (Theme 4), although there were certain restrictions and hygiene concepts in place, which formed the fifth theme regarding the interim terms of use. In their narrative review on the health benefits and risks of playing golf during the COVID-19 crisis, Robinson et al. argued that physical activity, while playing golf, is important for golfer´s fitness and mental health, and outweighs the risk of potential virus transmission, which is considered as being extremely low in outdoor environments, as aforementioned [31]. Thus, measures of well-being such as sense of belonging and life satisfaction significantly enhanced after golfing restrictions were relaxed in U.K. golfers [29].

In this verve, positive feelings such as sense of belonging and the social aspects of the club sport golf was summarized by the Theme 6, namely "Golf as a kind of religion". Therein, the survey participants very clearly expressed that they took their sports seriously and missed practicing at the golf course. The special features of golf make this sport unique, and athletes often practice it for decades, even if physical strength decreases with disabilities or age, as shown, e.g., by Stenner and co-workers in their paper on motives for the elderly to play golf [3]. The sense of belonging is often higher than in other sports, which aligns with Siegenthaler and O'Dell, who proposed four types of golfers regarding their level of seriousness about golf [7]. For the core devotees, golf is a central focus of their lives; moderate devotees highlight their enjoyment of golf and all associated activities; for social golfers, golf is a means for meeting and interacting with people; and, finally, therapy players think that golf helps them manage physical and emotional challenges. It seems that the degree of seriousness about the sport as a proxy for physical and cognitive involvement is important for successful aging. This connection between seriously playing golf and physical and mental benefits of an active lifestyle might be a further explanation for the higher life expectancy in golfers shown in the Swedish cohort study [6]. In agreement with the findings of life satisfaction levels in golfers, the high rate of open question responses in this study as an indirect measure for the degree to which individuals are affected by the crisis were reasonable [4,27,29].

In the second section of the findings from this qualitative content analysis, 42% of respondents named at least one alternatively practiced sport, with low-equipment outdoor sports such as hiking, biking, walking, and running being the most popular ones. This observation is in accordance with previous reports on the reminiscence of outdoor nature for physical activities due to the COVID-19 crisis [23,40,41]. This might not be purely voluntarily but triggered and forced by the sudden closure of indoor and outdoor sport facilities, and also other venues for cultural and social life. Additionally, the fear of contracting the virus and the need to maintain social distancing can also discourage people from engaging in physical activity. Reducing physical activity during the COVID-19 crisis can have significant negative effects on public health. For example, people who are less active are at an increased risk for chronic diseases, which can lead to a decrease in the quality of life. Moreover, the inactivity during the pandemic can also cause weight gain, which can be difficult to reverse. To address this public health issue, it is important for

policy makers and public health officials to provide guidance and support to help people maintain regular physical activity during the pandemic. This can include providing safe and accessible public spaces for physical activity, encouraging people to engage in outdoor activities, and promoting alternative forms of physical activity that can be performed at home. Additionally, providing education and resources on how to stay active while following COVID-19 guidelines would be useful.

From the golfers' perspective, the limited options of playing their favorite sport in combination with curfews, home office, closing of restaurants, theaters, and cancellation of sporting events eventually freed up resources for sports. As frequently mentioned by the participants, golf is their main leisure time activity, and they would prefer playing golf rather than engaging in alternative sports. This observation is in line with other studies on golfers, arguing that golf adds higher health and well-being benefits compared to other leisure activities with active golfers [4,29,31]. These benefits are potentially caused by the unique features of golf, including being outdoors while playing, thus experiencing nature and benefiting from its stress-reducing potential, being physically active regularly and for a longer time period of usually several hours, walking during tournaments, as well as engaging in social activities and community life [4,19,42,43]. So, the motivation for playing golf is multifaceted and certain aspects are weighted differently in every individual, depending on goals in the sport, and needs for personal well-being. The qualitative analysis of responses in our survey to an open-ended question on how the situation was experienced helped to shed light on circumstances and conditions influencing physical activity and life satisfaction during the pandemic. Our study found that a significant percentage of golfers surveyed expressed negative feelings about the impact of the pandemic on their ability to engage in their sport. Notably, it is difficult to compare this study's findings to similar studies regarding other sports, especially indoor sports, and other populations, such as elite athletes with less restricted access to sport facilities [44]. Better understanding of current conditions could potentially support organizational and political efforts to mitigate the negative effects that golfers and other athletes are experiencing during, and in the wake of this pandemic, or future ones.

*Limitations*

While this qualitative analysis sheds light on the perceived impact of the pandemic on golfers, the study findings should be contextualized in view of some methodological limitations. As in all qualitative studies, researcher bias potentially influenced findings. However, a third researcher who was not involved in the original coding checked and validated the analyses in the current study. Notably, the goal of qualitative research is to understand the experiences and perspectives of the participants, rather than to generalize findings to a larger population. Therefore, it is not necessary for all themes to be shared by all participants, as long as the themes that are identified are representative of the experiences and perspectives of the participants who do share them. It is also possible that some themes may be more prominent among certain subgroups of participants and less prevalent among others. For instance, age, gender, living conditions, health status, and income could influence the responses to the free-text boxes. To make the quotes more interesting and concrete, we mentioned the self-reported age and gender of each respondent. However, the quotes were only examples, as the data set was huge and a lot of quotes were actually summarized in a certain theme, as usually done in qualitative studies.

Our study sample was substantially larger than those of previous qualitative studies that explored the perceptions of golfers [3,7,45]. The online format of the survey limited participation to those with internet access and adequate technical skills, potentially introducing a selection bias. We collected self-reported data, which might lead to a recall bias. The study sample with an average age of about 55 years involved more males than females, and a predominantly aged population. As golf is widely regarded as a male-dominated sport of middle-aged men, this socio-demographic structure is commonly seen in studies including golfers [45]. Thus, we concluded that the participants in our study were very

likely to represent a cross-section of German-speaking golfers [29]. Future research using mixed-method and longitudinal study design is needed to elucidate longer-term effects on mental health caused by the COVID-19 pandemic in golfers.

## 5. Conclusions

The rationale behind this qualitative analysis of free-text comments of a large sample of German-speaking golfers in the context of the COVID-19 crisis was to get insight and to shed light on how people perceived the closure of outdoor sports facilities on the prominent example of golf. This is relevant from a public health perspective, as the value of nature experience and the extremely low virus infection risk in open spaces provided with large air ventilation is evident. Above this, golf is a sport that can be easily played without close social contact. Exploration of golfers' perceptions during the pandemic provided important insight, highlighting negative feelings for policy decisions concerning closure of outdoor sports facilities. Because of the positive aspects of regularly engaging in an outdoor sport, exercise should be encouraged during the pandemic as long as social distance is kept. Providing athletes with adequate information and safety instructions could be a concrete measure to help keeping them active in order to maintain their fitness level and enhance their life satisfaction and well-being.

**Author Contributions:** Conceptualization, D.J. and D.H.; data curation, D.J., S.G. and D.H.; formal analysis, D.J. and S.G.; investigation, D.J. and D.H.; methodology, D.J., S.G. and D.H.; resources, S.G. and D.H.; software, D.H.; validation, D.H.; writing—original draft, D.J., S.G. and D.H.; writing—review and editing, D.J., S.G. and D.H. All authors have read and agreed to the published version of the manuscript.

**Funding:** This research received no external funding.

**Institutional Review Board Statement:** The study was approved by the Ethics Committee of the Medical University of Vienna, Austria, on 1 April 2021 (protocol code 0505522) and conducted according to the guidelines of the Declaration of Helsinki.

**Informed Consent Statement:** Written online informed consent was obtained from all subjects involved in the study.

**Data Availability Statement:** The data supporting the findings of this study are available from the corresponding author upon request.

**Acknowledgments:** The authors thank M. Angerer for his valuable assistance in collecting online survey data. The authors are indebted to the administrators of golf-related social media channels, Perfect Eagle Golf, the Austrian Golf Association, the German Golf Association, and other individuals for sharing the survey link. The authors also would like to express sincere appreciation to the participating golfers for sharing their experiences in challenging times.

**Conflicts of Interest:** The authors declare no conflict of interest.

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
