# Peer review of "Perceived Impact of the COVID-19 Pandemic on Playing Golf: A Qualitative Content Analysis Study"

_2673-947X, doi:10.3390/hygiene3010006_

Round 1
Reviewer 1 Report
Dear authors,
I think that the manuscript highlights an important but less studied issue related to the COVID pandemic. In my opinion the manuscript should be published after minor revisions.
Abstract: the consequence of the concepts is not clear. In particular, the conclusions should be linked to the emergencial context and to the results.
Methods: delete the terms in line 146; the way by which the questionnaire was spread among golfers should be reported.
Discussion: the findings of the study should be discussed also in comparison to similar studies regarding other sports, especially indoor sports (see for example 10.3390/ijerph192013236).
Only the 42% practiced alternative activities. What about the others? They suspended every physical activity? Please discuss this as a possible public health issue.
Author Response
Reviewer 1
Comments and Suggestions for Authors
Dear authors,
I think that the manuscript highlights an important but less studied issue related to the COVID pandemic. In my opinion the manuscript should be published after minor revisions.
Authors’ response:
Dear Reviewer 1!
We thank you for the overall favorable evaluation of our research paper and the constructive feedback, which we incorporated in the revised version of the manuscript accordingly!
Comment:
Abstract: the consequence of the concepts is not clear. In particular, the conclusions should be linked to the emergencial context and to the results.
Authors’ response:
We agree and modified the last phrases of the abstract to the following, in order to highlight the context of the pandemic:
“In times of a pandemic, public spaces should be designed to allow for safe physical activity to maintain a mentally and physically fit population. In this context, policy makers should provide hygiene concepts that allow for minimal disturbance of sport routines, especially in regard to outdoor sports such as golf.”
Comment:
Methods: delete the terms in line 146; the way by which the questionnaire was spread among golfers should be reported.
Authors’ response:
In response to this request, we deleted the crossed-out text in line 146 and added the following information on how the survey link was spread in the methods section:
“In short, we distributed the online survey link via social networks Facebook and WhatsApp of several golf organizations, and golf clubs, as well as personal invitations.”
Comment:
Discussion: the findings of the study should be discussed also in comparison to similar studies regarding other sports, especially indoor sports (see for example 10.3390/ijerph192013236).
Authors’ response:
We agree. In response to this request, we modified discussion section, also adding the mentioned citation “Swimming at the Time of COVID-19: A Cross-Sectional Study among Young Italian Competitive Athletes”.
“Our study found that a significant percentage of golfers surveyed expressed negative feelings about the impact of the pandemic on their ability to engage in their sport. Notably, it is difficult to compare this study's findings to similar studies regarding other sports, especially indoor sports, and other populations, such as elite athletes with less restricted access to sport facilities.”
Comment:
Only the 42% practiced alternative activities. What about the others? They suspended every physical activity? Please discuss this as a possible public health issue.
Authors’ response:
In our study, we evaluated free text responses. However, not all free text responders mentioned alternative sport activities. We included this aspect in the discussion section, as requested.
“Additionally, the fear of contracting the virus and the need to maintain social distancing can also discourage people from engaging in physical activity. Reducing physical activity during the COVID crisis can have significant negative effects on public health. For example, people who are less active are at an increased risk for chronic diseases, which can lead to a decrease in the quality of life. Moreover, the inactivity during the pandemic can also cause weight gain, which can be difficult to reverse. To address this public health issue, it is important for policy makers and public health officials to provide guidance and support to help people maintain regular physical activity during the pandemic. This can include providing safe and accessible public spaces for physical activity, encouraging people to engage in outdoor activities, and promoting alternative forms of physical activity that can be done at home. Additionally, providing education and resources on how to stay active while following COVID-19 guidelines would be useful.”
Again, we thank you for your constructive feedback!
Reviewer 2 Report
Manuscript shows the qualitative data from golfer’s players during COVID-19 pandemic. I have some suggestions for you.
How golfers were contacted?
Are there any inclusion/exclusion criteria? I read the reference 25 but little information about that should be added.
The number of Decision and the date of approval of an Ethic Committee is usually stands at the end of the subsection Study Design.
2.1 section should be divided into study design and procedure or questionnaire.
Line 193: review punctuation.
Line 208: idem.
Results: Why is there different number of participants in each theme? I suppose that as a qualitative study, not all themes are shared for all participants, but this idea must be clarified.
Author Response
Reviewer 2
Comments and Suggestions for Authors
Manuscript shows the qualitative data from golfer’s players during COVID-19 pandemic. I have some suggestions for you.
Authors’ response:
Dear Reviewer 2!
We thank you for the overall favorable evaluation of our research paper and the constructive feedback, which we incorporated in the revised version of the manuscript accordingly!
Comment:
How golfers were contacted?
Authors` response:
We agree that information on how we contacted the participants is important, and is now given as follows:
“In short, we distributed the online survey link via social networks Facebook and WhatsApp of several golf organizations, and golf clubs, as well as personal invitations.”
Comment:
Are there any inclusion/exclusion criteria? I read the reference 25 but little information about that should be added.
The number of Decision and the date of approval of an Ethic Committee is usually stands at the end of the subsection Study Design.
2.1 section should be divided into study design and procedure or questionnaire.
Line 193: review punctuation.
Line 208: idem.
Authors` response:
We thank you for this valuable feedback. We now describe the inclusion/exclusion criteria in more detail:
“Data sets of German-speaking adults that played golf, as indicated by a filter question, and provided free text comment in the online survey were included in this qualitative analysis.”
Further, we renamed the 2.1 section into study design and questionnaire, and divided those aspects visually in two distinct small paragraphs. Also, we moved the date of approval of an Ethic Committee to the end of the part on Study Design.
We also corrected the punctuation issues.
Comment:
Results: Why is there different number of participants in each theme? I suppose that as a qualitative study, not all themes are shared for all participants, but this idea must be clarified.
Authors` response:
Yes, this is a normal finding in a qualitative study that the number of participants vary across different themes or categories. This is because the goal of qualitative research is to understand the experiences and perspectives of the participants, rather than to generalize findings to a larger population. Therefore, it is not necessary for all themes to be shared by all participants, as long as the themes that are identified are representative of the experiences and perspectives of the participants who do share them. It's also possible that some themes may be more prominent among certain subgroups of participants and less prevalent among others.
In response to this request, we added the following explanation to the limitations section to increase clarity:
“Notably, the goal of qualitative research is to understand the experiences and perspectives of the participants, rather than to generalize findings to a larger population. Therefore, it is not necessary for all themes to be shared by all participants, as long as the themes that are identified are representative of the experiences and perspectives of the participants who do share them. It is also possible that some themes may be more prominent among certain subgroups of participants and less prevalent among others.”
Again, we thank you for the evaluation of our research paper and the constructive feedback!
Reviewer 3 Report
This is a clear and well-written manuscript assessing the factors and situations that affected golfers' responses to closures for COVID-19 in German-speaking countries. The content is original, the statistical analysis is correct, and the sample size is large.
In general, the article may be of interest to sports scientists and policymakers.
I have only two suggestions for improving the article:
1- In the Introduction, it is appropriate to expand on the more general issue related to physical activity and public urban green space utilization opportunities during COVID-19 (see for example doi: 10.1007/s11524-017-0167-9 and doi: 10.3390/ijerph19159248 ).
2- I suggest expanding the discussion on the limitations of the present study. In your analysis, you considered the influence of golfers' sex and age but not their socioeconomic status. Instead, it appears from the literature that this is an important factor (see https://www.esri.ie/system/files/publications/RS63_0.pdf )
Author Response
Reviewer 3
Comments and Suggestions for Authors
This is a clear and well-written manuscript assessing the factors and situations that affected golfers' responses to closures for COVID-19 in German-speaking countries. The content is original, the statistical analysis is correct, and the sample size is large.
In general, the article may be of interest to sports scientists and policymakers.
Authors’ response:
Dear Reviewer 3!
We thank you for the overall favorable evaluation of our research paper and the constructive feedback, which we incorporated in the revised version of the manuscript accordingly!
Comment:
I have only two suggestions for improving the article:
1- In the Introduction, it is appropriate to expand on the more general issue related to physical activity and public urban green space utilization opportunities during COVID-19 (see for example doi: 10.1007/s11524-017-0167-9 and doi: 10.3390/ijerph19159248 ).
Authors´ response:
We agree and have now expanded the intro section, as suggested. We added the paper by Grigoletto et al.: Physical Activity Behavior, Motivation and Active Commuting: Relationships with the Use of Green Spaces in Italy.
“Grigoletto et al. found that during the Covid-19 pandemic, people visited urban parks in order to relax and to engage in physical activity in an Italian sample [24]. As a re-sult, green spaces and inner-city parks were heavy-used for recreational purposes and became overcrowded, as shown in a study from Vienna.”
Comment:
2- I suggest expanding the discussion on the limitations of the present study. In your analysis, you considered the influence of golfers' sex and age but not their socioeconomic status. Instead, it appears from the literature that this is an important factor (see https://www.esri.ie/system/files/publications/RS63_0.pdf )
Authors´ response:
We totally agree, that socioeconomic status is an important status in analysis, even if golfers are more prone to be of higher socioeconomic status than the general population. As for the collected information, we assessed sociodemographic characteristics age (in years), gender (male, female, or divers), and education level (primary, secondary, or tertiary), country of residence (Austria, Germany, or other countries), golf experience (0-5 years, 6-10 years, or 11 years or longer), and region (urban or rural). The results can be found in Table 1 and the results section. To make the quotes more interesting and concrete, we mentioned the self-reported age and gender of each respondent. However, the quotes were only examples, as the data set was huge and a lot of quotes were actually summarized in a certain theme, as usually done in qualitative studies.
We explained this aspect, among others, in the limitations section:
“Notably, the goal of qualitative research is to understand the experiences and perspectives of the participants, rather than to generalize findings to a larger population. Therefore, it is not necessary for all themes to be shared by all participants, as long as the themes that are identified are representative of the experiences and perspectives of the participants who do share them. It is also possible that some themes may be more prominent among certain subgroups of participants and less prevalent among others. For instance, age, gender, living conditions, health status, and income could influence the responses to the free text boxes. To make the quotes more interesting and concrete, we mentioned the self-reported age and gender of each respondent. However, the quotes were only examples, as the data set was huge and a lot of quotes were actually summarized in a certain theme, as usually done in qualitative studies.”
Again, we thank you for the evaluation of our research paper and the constructive feedback!